# Formulation of substrates with agricultural and forestry wastes for Camellia oleifera Abel seedling cultivation

Fei Zhou[1☯], Nianjin Wang[2☯], Jinping Zhang[1]*, Xiaohua Yao[1], Tiantian Zhang[1], Xiaofeng Zhang[3], Lingyan Zhan[4], Jieman Li[4]

1 Research Institute of Subtropical Forestry, Chinese Academy of Forestry, Hangzhou, Zhejiang, China, 2 Chun 'an County Forestry Bureau, Hangzhou, Zhejiang, China, 3 Fuyang District Environmental Protection Bureau, Hangzhou, Zhejiang, China, 4 Taihu County Huayuan Agricultural Science and Technology Development Co. LTD, Anqing, Anhui, China

☯ These authors contributed equally to this work.
* jinpingzhang@126.com

**Data Availability Statement:** All relevant data are within the paper and its Supporting Information files.

## Abstract

Five Camellia oleifera Abel seedling substrates were prepared using the conventional formula, but with the peat substituted by the composts of Camellia oleifera shell, pine chips, palm fiber residues, chicken manure, and sheep manure. The physical and chemical properties of the prepared substrates before and after seedling cultivation were determined and their effects on the growth of Camellia oleifera seedling were analyzed. It was found that the survival rates of the one-year-old seedlings produced from stem cuttings on all substrates at 6 months were greater than 97.5%. As compared with the conventional substrate, the substrates formulated with the composts were able to promote the seedling growth based upon height, ground diameter, root length and root volume measurement. The substrate prepared with the compost of Camellia oleifera shell+ palm fiber residue+ chicken manure (A3), vermiculite and perlite (6:3:1) was the most optimal, which gave 100% seedling survival rate, the greatest seedling height, and the largest ground diameter. In particular, the ground diameters and 26.67% of the seedling heights reached the grade 1 standard for two-year-old seedlings.

## 1. Introduction

In China, 2 billion tons of a variety of agricultural and forestry wastes are produced every year. These wastes are renewable and biodegradable biomasses, yet only a small portion is reused. The majority of them are incinerated, buried, or disposed randomly, which is the waste of resources and causes environmental pollutions [1–3]. Agricultural and forestry wastes can be reutilized as the cultivation substrates for flowers, seedlings, and vegetables after being treated by high-temperature aerobic fermentation [2]. Such applications not only re-utilize biomass wastes and reduce environmental pollutions, but also effectively improve the ecological environment [4] and realize carbon sequestration.

Camellia oleifera Abel. is an important woody oil tree species in southern China [5]. The rapid development of tea tree oil processing industry has dramatically increased the

**Funding:** The authors are grateful for the financial support from the National Key R&D Program of China (Grant No. 2019YFD1001602) and the Provincial Department of Science and Technology of Zhejiang, China (Grant NO.2017C02022).But the funders had no role in study design, data collection and analysis, decision to publish, or preparation of the manuscript.

**Competing interests:** The authors have declared that no competing interests exist.

production of by-products, such as Camellia oleifera shells. Composting these by-products can not only prevent wasting resources and environmental pollution, but also provide substrates and organic fertilizers to improve soil quality and inhibit soil-borne diseases. Pot method with light substrates shows great advantages, such as all-season operation, high survival rates, no slow growth period, fast growth, less root damage during transplanting, long afforestation seasons, and low transportation costs for seedling production [6–9]. Substrate is one of the key factors affecting the performance of pot production of seedling [10]. The conventional Camellia oleifera seedling substrates are formulated with topsoil and peat. Peat is a non-renewable resource, and its performance for Camellia oleifera seedling production can be further improved [11]. Substituting peat with agricultural and forestry wastes has become a research hotspot. For example, Lu et al. reported the Camellia oleifera seedling production using the substrate formulated with the compost of forest wastes, weeds, and vermicompost [3]. Wu et al. successfully obtained Camellia oleifera seedlings with the substrate containing the compost of sawdust, decomposed litter, biogas residue and yellow soil [11]. Dai et al. also formulated the substrate with the compost of bagasse, cassava skin, peanut shells, charcoal ash, and garden soil to produce Camellia oleifera seedlings [12]. All these reports used the composts of agricultural and forestry wastes as the substrates for the seedling production of Camellia oleifera, yet lacked the detailed information of the raw material sources, composting formula and composting method. The variation in the substrate raw material source makes it difficult to obtain consistent composts. The quality of Camellia oleifera shell compost has been successfully improved by adding different nitrogen sources, such as urea, compound fertilizer, and pig manure and EM bacteria [13], but no application research on the composts has been carried out.

Composting is a major process of recycling agricultural and forestry wastes. It is generally divided into aerobic composting and anaerobic composting. Aerobic composting decomposes organic matters under aerobic conditions mostly into $CO_2$ and water and releases heat. Anaerobic composting mainly produces methane, $CO_2$, and low molecular weight intermediates, such as organic acids, etc. Most conventional composting is conducted under anaerobic conditions, which tends to release odors [14]. The modern composting processes are mainly conducted under aerobic conditions with controlled water contents, C/N ratios and ventilation. Aerobic composting can transform unstable agricultural and forestry wastes into stable substances, during which pathogenic bacteria are killed. The composts thus can be safely handled and stored. They are also a good seedling substrate, soil conditioner and organic fertilizer [15].

In this work, aerobic composting was conducted with Camellia oleifera shells, pine chips, palm fiber residues, chicken manure and sheep manure as the main raw materials using different formulas. The obtained composts were respectively mixed with boiler slag, vermiculite, and perlite at certain ratios as the substrates to grow Camellia oleifera seedlings. The changes in the physical and chemical properties of the substrates after seedling cultivation and the relationship between composting raw materials and the seedling development were comprehensively evaluated and analyzed, aimed to optimize a composting formula and substrate formula for the cultivation of Camellia oleifera seedling and provide experimental data and a scientific reference for the recycling and composting of agricultural and forestry wastes including Camellia oleifera shells as the substrate exclusively for the pot cultivation of Camellia oleifera seedlings.

## 2. Materials and methods

### 2.1 Compost materials and preparations

Camellia oleifera shells, palm fiber residues, chicken manure, pine chips, and sheep manure were provided by Anhui Wanxiuyuan Ecological Agriculture Group Co., Ltd., (Anqing, Anhui

**Table 1. Primary properties of compost raw materials.**

| Raw material/Property | TOC (%) | TN (%) | C/N | Cellulose/% | Hemicellulose/% | Lignin/% |
|---|---|---|---|---|---|---|
| Camellia oleifera shells | 51.80±2.58 | 0.729±0.03 | 71.078±3.09 | 18.62±0.22 | 49.34±0.07 | 29.71±0.14 |
| Brown silk chips | 46.80±2.23 | 2.68±0.14 | 17.506±1.17 | 28.15±0.18 | 20.58±0.08 | 44.08±0.16 |
| Pine chips | 54.20±2.74 | 0.285±0.01 | 190.719±12.52 | 48.36±0.11 | 27.59±0.12 | 19.58±0.21 |
| Chicken mature | 34.39±1.75 | 2.73±0.14 | 12.667±0.58 | - | - | - |
| Sheep mature | 36.47±1.81 | 2.78±0.14 | 13.186±0.21 | - | - | - |
| Urea | 20±0.18 | 46.4±1.22 | 0.431±0.01 | - | - | - |

"-"indicates extremely low content or none

Province, China). The Camellia oleifera shells and pine chips were smashed to the sizes of 5–8 mm before use. The compost bacteria were purchased from Yijiayi Biological Engineering Co., Ltd. (Zhengzhou, China) and urea was obtained from Henan Jinkai Chemical Investment Holding Group Co., Ltd (Zhengzhou, China). The primary properties of the compost raw materials are listed in Table 1.

## 2.2 Composting process

Composting was conducted by window composting in the composting plant of Anhui Wanxiuyuan Ecological Agriculture Group Co., Ltd. from August to November 2020. Four composting piles including A1: Camellia oleifera shell + urea (C/N 60); A2: Camellia oleifera shell + sheep manure (C/N 55); A3: Camellia oleifera shell + palm fiber residue + chicken manure (C/N 30); and A4: pine chips + chicken manure (C/N 40.7) were prepared, mixed, and stirred evenly. The moisture content of each pile was adjusted to 55–60% and 0.1% fermentation bacteria were added into each pile. Each pile was then thoroughly mixed, stirred evenly, and stacked for high-temperature aerobic fermentation at the same time. The composting piles were turned for aeration and sampled by the five-point sampling method every 7 days. For each sample, half portion was stored at -20˚C and the other half was dried at 65˚C and ground.

## 2.3 Compound substrate preparation and seedling cultivation

Seedlings were cultivated at the nursery of Anhui Wanxiuyuan Ecological Agriculture Group Co., Ltd. located in Taihu County, Anqing, Anhui Province between 30˚09' N to 30˚46' N and 115˚45 'E to 116˚30 'E. The temperatures during the experiment varied between from -6˚C and 34˚C, and there were 24 days with temperatures lower than 0˚C.

Compound substrates were prepared with the composts and inorganic matrices using the formulas shown in Table 2. In December 2020, one year old Camellia oleifera seedlings with intact root systems, the heights of 13 ± 3 cm and ground diameters of 0.22 ± 0.02 cm were respectively transplanted into pots with the diameter of 16 cm and height of 16 cm. The pots

**Table 2. Formulas of compound substrates.**

| Treatments | Substrate formula |
|---|---|
| A | (Camellia oleifera shell + urea):Vermiculite: Perlite = 6:3:1 |
| B | (Camellia oleifera shell + sheep manure):Vermiculite: Perlite = 6:3:1 |
| C | (Camellia oleifera shell + palm fiber residue+chicken manure):Vermiculite: Perlite = 6:3:1 |
| D | (Camellia oleifera shell + palm fiber residue+chicken manure):Boiler slag:Vermiculite: Perlite = 4:2:3:1 |
| E | (Pine chip + urea):Vermiculite: Perlite = 6:3:1 |

were filled with the compound substrates listed in Table 2. For each substrate, 200 seedlings were planted and allowed to grow for 6 months. The substrates were then sampled and analyzed for physical and chemical properties. During the seedling cultivation, the substrates were maintained moist, and watered thoroughly when they became dry during the fast growing period.

## 2.4 Analytical methods

During composting, the compost temperature was measured from the upper (10 cm from the top), middle and lower (10 cm from the bottom) parts of the pile at around 3:00 pm every day, and the ambient temperature of the day was recorded at the same time. The bulk density, total porosity, aeration porosity, water-holding porosity, electrical conductivity (EC) and pH were also measured following the Chinese Forestry Industry Standard GB/T 33891–2017 of organic substrates for greening [16]. Total organic carbon (TOC), total nitrogen (TN), total phosphorus (TP), total potassium (TK) and germination index (GI) were determined by the method of Zhang et al. [17]. The organic matter content, total nutrient content, C/N ratio and survival rate of Camellia oleifera seedling were calculated with Eq (1), (2), (3) and (4), respectively.

$$\text{Organic matter } (\%) = \text{TOC } (\%) \times 1.724 \tag{1}$$

$$\text{Total nutrient } (\%) = \text{TN } (\%) + P_2O_5 \ (\%) + K_2O \ (\%) \tag{2}$$

$$C/N = \text{TOC}(\%)/\text{TN}(\%) \tag{3}$$

$$\text{Survival rate} = \text{number of survived seedlings} \div \text{total number of planted seedlings} \\ \times 100\% \tag{4}$$

After 185 days of cultivation, 60 seedlings were randomly collected from each group and measured for ground diameter and height. Five seedlings were randomly selected from each group and analyzed for root length, surface area, volume and average diameter using a Wanshen LA-S root system analyzer to evaluate the effects of the physical and chemical properties of substrate on the growth of Camellia oleifera seedling. The obtained data were processed and analyzed using the EXCEL and IBM SPSS Statistics.

## 3. Results and discussion

### 3.1 Compost temperature

Temperature is one of the important indicators to evaluate the compost maturity [18, 19]. A composting process is generally divided into four phases, e.g., the mesophilic phase, the thermophilic phase (>50°C), the cooling phase, and the maturation phase. Pathogenic microorganisms, insect eggs and weed seeds are killed at the thermophilic phase when the compost temperature increases to over 55°C, and thus compost becomes harmless. The fermentation bacteria are rapidly deactivated as temperature increased to over 63°C, and the compost temperature drops, leading to cooling phase [20]. As can be seen from Fig 1, the temperatures of all compost piles are much higher than the ambient temperature during the composting, indicating that ambient temperature has little effects on the composting. The initial temperatures of all composting piles were 27±0.6°C. The temperatures of A1, A2, A3 and A4 increased to above 50°C on the day 6, 7, 3 and 3, respectively, and remained at 50°C for 20, 23, 39, and 9 days, and at 60°C for 4, 7, 22, and 6 days, respectively, conforming to the safety and hygiene requirements [21]. The four compost piles entered the cooling phase on day 33, 34, 44 and 15,

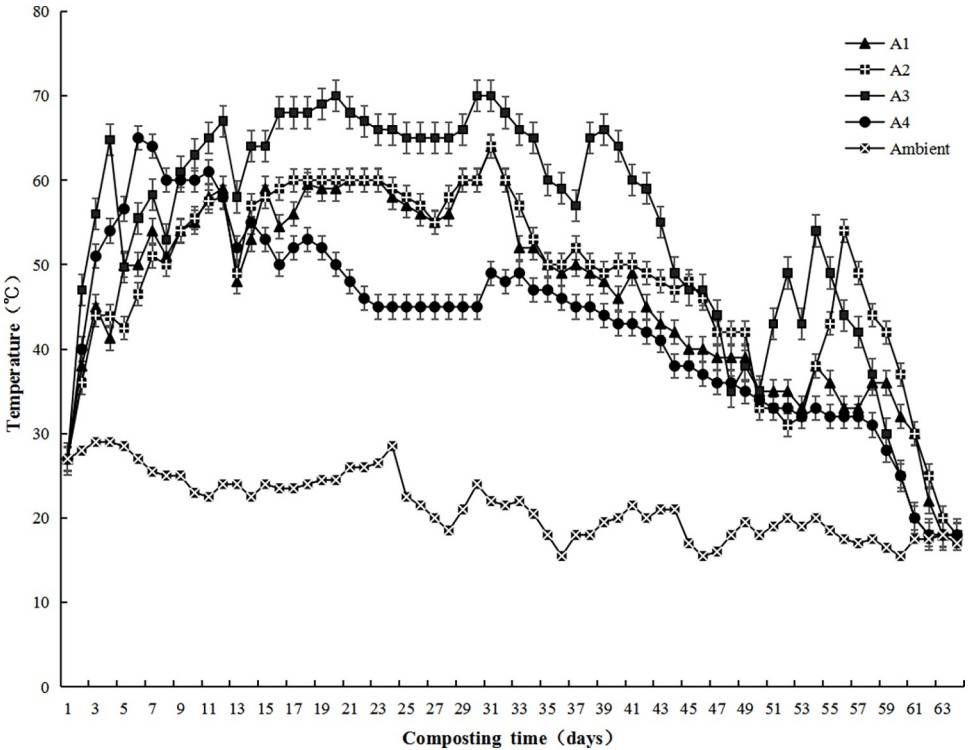

**Fig 1. Temperature changes of the four compost piles during composting.** We have uploaded the minimum data set of the above chart (except Table 2) as a separate S1 Data.

respectively and their maturation phases lasted 38, 42, 44 and 21 days, respectively. The sudden temperature drops are due to heat dissipation during manual turning for aeration [22, 23].

The compost pile A3 exhibited the largest heating rate, the highest high temperature, and the longest thermophilic phase. It can be explained that, first, the initial C/N (30) of A3 is most suitable for microbial reproduction [20]. Second, there is a large amount of chicken manure in the pile. Chicken manure is rich in protein and urea that can be easily decomposed and utilized by microorganisms. Third, the palm fiber residues can improve the aeration porosity of the compost, which is conducive to the proliferation of the microorganisms and heat generation. Piles A1, A2 and A3 all contain Camellia oleifera shells, but have different C/N ratios due to different excipients. The C/N ratio of A3 is 30, which is in the optimal range (25–35), while those of A1 and A2 are 60 and 55, respectively, that indicate low nitrogen contents. The activity of microorganisms is lower under the insufficient nitrogen conditions, and thus the composting reactions are slow [24]. The temperature of pile A4 increased rapidly and thus its initial temperature was higher. However, it entered the cooling phase earliest and its temperature dropped fastest. It can be explained that the nutrients in the chicken manure are rapidly decomposed. The main components in pine chip are cellulose and lignin that are difficult to be degraded and the hemicellulose content is only 27.59%. In contrast, the Camellia oleifera shells in other three groups are rich in highly degradable hemicellulose (49.34%), which can prolong the thermophilic phase. These results suggest that the physical and chemical properties of raw materials directly affect the composting process and the physical and chemical properties of compost (Table 3).

**Table 3. Physical and chemical properties of the composts of agricultural and forest waste and other components for the preparation of compound substrates.**

| Group | Moisture content MC/ % | bulk density g·cm⁻³ | Total porosity % | Aeration porosity % | Water-holding porosity % | pH | EC (ms·cm⁻¹) |
|---|---|---|---|---|---|---|---|
| A1 | 10.27±0.52 | 0.39±0.018 | 73.94±3.66 | 6.07±0.29 | 67.87±3.39 | 7.00±0.21 | 0.13±0.02 |
| A2 | 9.46±0.48 | 0.40±0.021 | 74.13±3.68 | 9.30±0.47 | 64.83±3.23 | 8.00±0.26 | 0.08±0.03 |
| A3 | 13.82±0.68 | 0.47±0.024 | 65.90±3.30 | 11.83±0.60 | 54.07±2.71 | 7.79±0.24 | 0.20±0.01 |
| A4 | 22.47±1.11 | 0.42±0.02 | 83.40±4.15 | 1.80±0.089 | 81.60±4.07 | 7.00±0.22 | 0.06±0.03 |
| **Group** | **TN%** | **TK%** | **TP%** | **TOC%** | **total nutrient %** | **GI%** | **organic matter %** |
| A1 | 1.07±0.05 | 1.83±0.09 | 0.37±0.02 | 38.20±1.92 | 3.27±0.15 | 89.12 ±4.44 | 65.86±3.25 |
| A2 | 1.16±0.057 | 2.07±0.12 | 0.62±0.03 | 36.30±1.82 | 3.85±0.21 | 90.13 ±4.49 | 62.58±3.11 |
| A3 | 2.75±0.14 | 2.55±0.11 | 1.02±0.05 | 29.80±1.48 | 6.32±0.33 | 99.42 ±0.80 | 51.38±2.54 |
| A4 | 0.84±0.04 | 0.54±0.03 | 0.42±0.02 | 47.30±2.37 | 1.80±0.08 | 91.31 ±4.52 | 81.55±4.06 |
| **Raw materials** | **Moisture content MC/ %** | **bulk density g·cm⁻³** | **Total porosity %** | **Aeration porosity %** | **Water-holding porosity %** | **pH** | **EC(ms·cm⁻¹)** |
| Boiler slag | 2.08±0.11 | 0.70±0.03 | 63.10±3.16 | 0.40±0.019 | 62.7±3.14 | 8.00 ±0.025 | 0.22±0.012 |
| Vermiculite | 1.55±0.078 | 0.13±0.01 | 95.04±4.75 | 30.03±1.51 | 65.01±3.24 | 7.00 ±0.021 | 0.047±0.0024 |
| perlite | 0.14±0.007 | 0.16±0.01 | 60.33±3.03 | 29.51±1.48 | 30.82±1.55 | 7.00 ±0.019 | 0.022±0.0011 |

## 3.2 Physical and chemical properties of composts and substrates

The composts of agricultural and forestry wastes can be used alone as a seedling substrate or can be formulated with other matrices to form different seedling substrates. The substrates with large bulk densities are favorable for the seedling handling. The cohesions of the substrates with small bulk densities are poor, which is inconducive to the root fixation [25]. The bulk densities of A1, A2, A3, and A4 composts are 0.39 g·cm⁻³, 0.40 g·cm⁻³, 0.47g·cm⁻³, and 0.42 g·cm⁻³, all within the ideal bulk density range of 0.1–0.8 g·cm⁻³ [26]. Porosity is another important physical property that affects the aeration, drainage and water holding capacity of a substrate. The total porosities of A1, A2 and A3 composts are 73.94%, 74.13% and 65.90%, respectively, which are all within the ideal total porosity range of seedling substrate (40%-75%) [27, 28]. The total porosity of A4 compost (83.4%) is too high as a seedling substrate and can be adjusted with other matrices. The pHs, seed germination indexes (GIs), and organic matter contents of the composts are in the ranges of 7–8, 89.12–99.42, and 65.86–81.55%, respectively, which meet the standard NY/T525-2021 for organic fertilizer defined by the Ministry of Agriculture and Rural Affairs of the People's Republic of China [29] (pH 5.5–8.5, GI≥70%, and organic matter content≥30%). Germination index (GI) is a parameter that can be used to quickly evaluate the phytotoxicity of a compost in a short period of time and is of great importance in actual productions. The GIs of all composts are >85%, and thus the composts can be considered non-toxic to plants [30]. The total nutrient contents of A1, A2, A3, and A4 composts are measured to be 3.27%, 3.85%, 6.32%, and 1.80%, respectively. That of A3 compost reaches the standard NY/T525-2021 [29] for organic fertilizer (≥4%) and thus it can be used as an organic fertilizer or a substrate component. The rest 3 groups can be used as cultivation substrates or substrate raw materials.

### 3.3 Changes in physical and chemical properties of compound substrates after seedling cultivation

The water holding capacities of the obtained composts are low, and thus vermiculite with a better water holding ability is introduced. The raw materials listed in Table 3 were mixed at the ratios listed in Table 2, and the physical and chemical properties of the corresponding compound substrates were measured (Table 4). The bulk densities of compound substrates A, B, C, D, and E were measured to be 0.41 g·cm$^{-3}$, 0.47 g·cm$^{-3}$, 0.41 g·cm$^{-3}$, 0.42 g·cm$^{-3}$, and 0.41 g·cm$^{-3}$, respectively and their total porosities were 73.83%, 67.83%, 66.70%, 68.17% and 74.60%, respectively, all within the ideal ranges of 0.1–0.8 g·cm$^{-3}$ and of 40%-75% [26–28]. Their aeration porosities (29.31%, 28.12%, 25.44%, 24.22% and 23.10%), pH (7.03–7.43) and EC values (0.12–0.20 ms·cm$^{-1}$) all conform to the national standard "Organic Substrates for Greening" (GB/T 33891–2017) of the People's Republic of China with the requirements [16] of total porosity≥20%, pH 5.0–7.6 and EC≤0.65 ms·cm$^{-1}$. Yet none of the pHs conforms to the Forestry Industry Standard LYT 2314–2014 "Technical Regulations for Seed Box Production of Camellia Container Seedling" [31] of the People's Republic of China (pH 5.0–6.5).

The changes in the physical and chemical properties of the five compound substrates after the seedling cultivation are summarized in Tables 4 and 5. As can be seen, the bulk densities, aeration porosities and EC values of all groups increase, while their total porosities and water-holding porosities, organic matter contents and total nutrient contents decrease after the 6-month cultivation. It can be explained that, first, most of the hemicellulose in the compost raw material is degraded during composting [32], and large amounts of refractory cellulose and lignin continue to decompose slowly during the cultivation, which increases the aeration porosity. Second, with the seedling growth and development, the root elongation squeezes the substrate and thus decreases the total porosity and water-holding porosity. Third, the seedling growth consumes large amounts of organic matters, nutrients, and minerals, which decreases the contents of organic matters, nitrogen, $P_2O_5$, $K_2O$ and total nutrient. The seedling development also accelerates the decomposition of refractory substances, resulting in increased EC

**Table 4. Changes in the physical and chemical properties of compound substrates after seedling cultivation.**

| group | | A | B | C | D | E |
|---|---|---|---|---|---|---|
| bulk density g/cm$^3$ | before | 0.41±0.02 | 0.47±0.02 | 0.41±0.02 | 0.42±0.02 | 0.41±0.02 |
| | after | 0.57±0.027 | 0.76±0.037 | 0.59±0.028 | 0.59±0.029 | 0.60±0.031 |
| | Increase% % | 39.02±4.429 | 61.70±2.73 | 43.90±1.761 | 40.48±3.14 | 46.34±0.896 |
| total porosity % | before | 73.83±3.56 | 67.83±3.25 | 66.70±3.21 | 68.17±3.26 | 74.60±3.53 |
| | after | 58.07±2.55 | 58.80±2.55 | 61.40±2.60 | 62.52±2.05 | 59.60±1.88 |
| | decrease | 21.35±0.36 | 13.31±0.54 | 7.95±1.38 | 8.29±1.63 | 20.11±2.01 |
| aeration porosity % | before | 29.31±1.35 | 28.12±1.31 | 25.44±1.22 | 24.22±1.19 | 23.10±1.12 |
| | after | 52.11±2.33 | 54.21±2.34 | 51.91±2.19 | 44.22±2.16 | 42.18±2.02 |
| | increase | 77.79±0.31 | 92.78±0.85 | 104.05±3.60 | 82.58±0.79 | 82.60±1.29 |
| Water-holding porosity % | before | 44.52±2.21 | 39.71±1.94 | 41.26±1.99 | 43.95±2.08 | 51.50±2.42 |
| | after | 5.96±0.22 | 4.60±0.21 | 9.49±0.42 | 18.30±0.37 | 17.41±0.83 |
| | decrease | 86.61±0.20 | 88.42±0.05 | 77.00±0.17 | 81.11±1.44 | 66.19±2.74 |
| pH | before | 7.43±0.29 | 7.38±0.24 | 7.03±0.26 | 7.34±0.27 | 7.20±0.23 |
| | after | 7.32±0.34 | 7.45±0.36 | 7.65±0.37 | 7.48±0.36 | 7.61±0.37 |
| | increase% | -1.48±1.125 | 0.95±1.612 | 8.82±1.24 | 1.91±1.211 | 5.69±2.28 |
| EC (ms·cm$^{-1}$) | before | 0.15±0.002 | 0.15±0.002 | 0.20±0.002 | 0.12±0.003 | 0.12±0.003 |
| | after | 0.46±0.023 | 0.22±0.012 | 0.24±0.007 | 0.18±0.005 | 0.24±0.002 |
| | increase% | 206.67±12.605 | 46.00±5.899 | 20.00±2.769 | 50.00±0.878 | 50.00±3.09 |

**Table 5. Changes in the nutrient contents of compound substrates after seedling cultivation.**

| group | | A | B | C | D | E |
|---|---|---|---|---|---|---|
| organic matters % | before | 54.33±2.72 | 50.90±2.53 | 43.00±2.13 | 37.00±1.82 | 61.50±3.06 |
| | after | 9.45±0.46 | 19.14±0.94 | 14.43±0.73 | 9.93±0.48 | 15.76±0.77 |
| | decrease | 82.61±1.36 | 62.40±2.24 | 66.44±3.35 | 73.16±2.50 | 74.37±2.53 |
| N% | before | 0.86±0.041 | 0.93±0.042 | 2.29±0.09 | 1.38±0.065 | 0.85±0.039 |
| | after | 0.49±0.022 | 0.51±0.023 | 0.68±0.032 | 0.65±0.031 | 0.24±0.011 |
| | decrease | 43.02±1.48 | 45.16±2.18 | 70.31±2.08 | 52.90±2.72 | 71.76±2.11 |
| $P_2O_5$% | before | 0.36±0.017 | 0.56±0.026 | 0.90±0.042 | 0.59±0.027 | 0.62±0.029 |
| | after | 0.095±0.0045 | 0.16±0.0079 | 0.58±0.025 | 0.22±0.0099 | 0.069±0.0032 |
| | decrease | 73.61±1.12 | 71.43±2.17 | 35.56±1.33 | 62.71±2.59 | 88.87±1.05 |
| $K_2O$% | before | 1.47±0.074 | 1.68±0.082 | 2.13±0.097 | 1.43±0.068 | 0.75±0.035 |
| | after | 0.59±0.027 | 0.72±0.034 | 0.63±0.031 | 0.61±0.029 | 0.43±0.021 |
| | decrease | 59.86±0.56 | 57.14±0.63 | 70.42±2.25 | 57.34±2.23 | 42.67±2.34 |
| Total nutrient % | before | 2.69±0.13 | 3.17±0.10 | 5.32±0.20 | 3.40±0.15 | 2.22±0.08 |
| | after | 1.18±0.05 | 1.40±0.05 | 1.89±0.03 | 1.47±0.06 | 0.75±0.02 |
| | decrease | 56.13±0.90 | 55.84±1.00 | 64.47±0.99 | 56.76±2.36 | 66.22±1.94 |

values. The seedlings of group C show the highest survival rate, the greatest seedling height, and the largest ground diameter (Table 6), and the aeration porosity of the substrate is increased by up to 104.05%. The total porosities of substrates A, B, C, D and E decrease by 21.35%, 13.31%, 7.95%, 8.29% and 20.11%, respectively, after the cultivation. Both substrates C and D contain the palm fiber residue a high lignin content (Table 1). Lignin is difficult to be degraded, which explains the smaller changes in the total porosity of two substrates. Therefore, it can be concluded that the substrate porosity changes after seedling cultivation are related to the cellulose and lignin contents of the compost feedstock. Substrates A, B, C and D all contain the Camellia oleifera shell with a high hemicellulose content (Table 1). The compost in substrate A was abstained in the absence of livestock or poultry manure and the microbial abundance [21] and temperature of the compost were lower than those of the compost in substrate B (Fig 1). Therefore, the hemicellulose degradation rate of substrate A is lower than that of substrate B [15]. The hemicellulose in substrate A was further degraded during the seedling cultivation, leading to the largest decrease in total porosity. The cellulose content of pine chips in substrate E is high (Table 1). Cellulose is difficult to be decomposed, but easier than lignin. Therefore, small amounts of hemicellulose, cellulose and lignin continue to decompose during the seedling cultivation, which causes the significant decreases in the total porosity of substrate E.

**Table 6. Morphologies of seedlings raised on different substrates.**

| Treatment | | A | B | C | D | E |
|---|---|---|---|---|---|---|
| survival rate % | | 99 | 99.5 | 100 | 97.5 | 99.5 |
| height/cm | average | 20.0±6.01 | 27.2±7.81 | 31.4±9.93 | 29.6±8.37 | 22.2±6.03 |
| | Maximum | 44.0 | 51.0 | 63.0 | 47.0 | 38.0 |
| Ground diameter/mm | average | 2.8±0.46 | 3.7±0.57 | 4.1±0.69 | 3.9±0.47 | 3.3±0.50 |
| | Maximum | 3.9 | 4.7 | 5.5 | 5.1 | 4.4 |
| total root length/cm | | 2169.79±107.96 | 4377.77±215.32 | 2161.92±107.21 | 2864.55±142.53 | 3900.97±192.67 |
| Root surface area/cm$^2$ | | 371.30±18.23 | 768.81±38.23 | 397.98±19.56 | 596.43±29.62 | 683.89±34.01 |
| Average root diameter/mm | | 0.56±0.026 | 0.56±0.027 | 0.58±0.027 | 0.66±0.032 | 0.56±0.025 |
| Root volume cm$^3$ | | 5.10±0.24 | 10.86±0.52 | 5.90±0.30 | 10.01±0.51 | 9.58±0.45 |

## 3.4 Relationship between physical and chemical properties of compound substrate and seedling morphology

Survival rate is an important indicator for evaluating the successful cultivation of seedlings. The seedling survival rates of all groups are above 97% (Table 6), significantly higher than that (42.50–67.50%) obtained using the compost of rice hulls, bark, and sawdust (rice hulls 20%-26%, bark 28%-44%, sawdust 36%-52%) as the substrate [8].

Seedling height and ground diameter are important indicators of seedling quality [26]. As shown in Table 6, the average heights and maximum heights of the seedlings cultivated on the 5 substrates for 6 months are in the order of C>D>B>E>A and C>B>D>A>E, respectively. Both of their average ground diameters and maximum ground diameters are in the order of C>D>B>E>A. The average ground diameter (4.1 mm) and average seedling height (31.4 cm) of group C respectively reach the first-grade and second-grade criteria defined in the standard GB/T 26907–2011 [33] for two years old seedling ($\geq$0.35 cm and $\geq$25 cm). In addition, 26.67% of the seedling heights reach the first-grade criteria ($\geq$40 cm), and 45% of them reach the second-grade criteria. The average ground diameters of both groups B (3.7 mm) and D (3.9 mm) reach the first-grade criteria. Upon their seedling heights, 6.7% of group B and 15% of group D reach the first-grade criteria and the rest all reach the second-grade criteria. The average ground diameter (3.3 mm) of the seedlings in group E meets the second-grade criteria ($\geq$0.30 cm), but only 28.3% of the seedling heights meet the second-grade criteria. Neither of the average ground diameter (2.8 mm) nor the average seedling height (20.0 cm) of the seedlings in group A conforms to the second-grade criteria, but 41.67% of the seedlings meet the second-grade standard for ground diameter and 15% of them meet the second-grade standard for height. It is worth noting that the average seedling height of the seedlings in group C (31.4 cm) is 30.5 cm higher than that of two-year-old Camellia oleifera seedlings cultivated on the substrate prepared with coconut bran and yellow soil (coconut bran 75% +yellow soil 25%) and 3 kg/m$^3$ of slow-release compound fertilizer from KOCH (USA) [6]. The average seedling heights (31.4 cm, 29.6 cm) and average ground diameters (4.1 mm, 3.9 mm) of the seedlings in groups C and D are better than those (28.70 cm, 3.93 mm) of the seedlings raised on the substrate formulated with peat and the compost of rice hulls (peat 50% + (compost of rice hulls + extruded perlite + vermiculite + boiler slag + imported slow-release fertilizer) 50%) for 18 months [34]. The average seedling heights (27.2 cm, 31.4 cm and 29.6 cm) and average ground diameters (3.7 mm, 4.1 mm and 3.9 mm) of the seedlings in groups B, C and D are better than those (25.98 cm, 3.43 cm) of the seedlings raised on the substrate containing composted bagasse, cassava skin and charcoal ash (2:1:1) for 18 months [12].

The compound substrate C contains the highest amounts of N, $P_2O_5$, $K_2O$, and total nutrients before seedling cultivation, and the decreases in the contents of N, $K_2O$, and total nutrient and the increase in aeration porosity are the most significant after the seedling cultivation (Tables 5 and 6). The seedling survival rate, height, and ground diameter of this group are also the greatest. These results suggest that seedling height and ground diameter are mainly related to the total nutrient content of the substrate, consistent with the results reported by Wu et al. and Zeng et al. [35, 36]. Compared with those of other substrates, the N and $K_2O$ contents of substrate C decrease most significantly, by 70.31% and 70.42% respectively, while the decrease in its $P_2O_5$ content is the smallest, suggesting that nitrogen and potassium can promote the growth and development of Camellia oleifera seedling [37–39].

The organic matter contents of substrates A-E decreased by 82.61%, 62.40%, 66.44%, 73.16% and 74.37%, respectively, after the seedling cultivation due to the consumption by the seedlings and losses during watering. Before the seedling cultivation, the organic matter contents and total nutrient contents of the substrates are in the order of E>A>B>C>D, and

C>D>B>A>E, respectively. The substrates with high organic matter contents generally show low total nutrient contents. The raw materials of the composts in substrates C and D are the same, but their formulas and ratios with other inorganic matrices are different. The ratio of the compost in substrate C is 20% higher than that in substrate D. Therefore, both the contents of organic matter and total nutrient of substrate C are higher than those of substrate D. The seedling heights and ground diameters of groups B, C, and D are significantly greater than those of the other two groups, but the decreases in the organic matter contents of their substrates are smaller, indicating that there are large amounts organic matters in the composts of Camellia oleifera shell and livestock/poultry manures that can be absorbed and utilized by the seedlings. The maximum decrease in the total nutrient content of substrate C is only 3.43%, and the average seedling height and ground diameter of the seedlings raised on it are the largest. Even though the organic matter content of substrate A decreased by 82.61% after the seedling cultivation, it showed no significant advantages for the seedling growth. It may be explained with the low degradation rate of lignocellulose during the composting that results in low contents of the organic matters usable for the seedling development. All these results suggest that the height and ground diameter of Camellia oleifera seedling are closely related to the properties of compost raw materials, composting formula, composting process and rate, and substrate formula. The stable compost raw materials, formula, composting process, and substrate formula are the keys to improving the quality and efficiency of the Camellia oleifera seedling cultivation.

The physical and chemical properties of substrate also affect the morphological characteristics of Camellia oleifera seedling root system. Root morphology can be characterized with root growth indicators, such as root length, root surface area, root volume, and root diameter [25, 40]. In general, the longer and thicker root systems with larger surface areas and volumes suggest better development and growth conditions. As shown in Table 6, the root morphologies of the seedlings raised on different substrates are significantly different. The seedlings of group B exhibit the longest root system of 4377.77 cm, the largest root surface area of 768.81 cm$^2$, and the largest root volume of 10.86 cm$^3$, and both the root surface area and volume of group A are the smallest. The root surface areas and root volumes of the seedlings raised on the five substrates are in the orders of B>E>D>C>A and B>D>E>C>A, respectively. The average root diameter of group D is 0.66 cm, significantly larger than those of other groups (0.56 cm for groups A, B and E, and 0.58 cm for group C). Despite the larger root length, surface area and volume of group B, its average root diameter is similar to those of other groups. Yet the overall root development of group B is the best, possibly because the porosity and nutrient content of its substrate is more suitable for the root development of Camellia oleifera seedling. The total root lengths, surface areas, and volumes of all five groups are better than those of the Camellia oleifera seedlings raised on conventional substrates. For example, the seedlings grown on the substrate formulated with 40% yellow soil + 15% pine forest topsoil + 20% mushroom residue + 20% peat + 5% manure for two years only show the average total root length of 280.32 cm, the surface area of 106.72 cm$^2$, and the average root volume of 8.71 cm$^3$ [10].

## 3.5 Correlation analysis of seedling morphology and substrate physical and chemical properties

Seedling height, ground diameter and root activity are generally considered to be the most important indicators of seedling quality [41, 42]. Herein, we evaluated the Camellia oleifera seedling growth with seedling height, ground diameter and root morphology and analyzed the correlation between the physical and chemical properties of substrate and the seedling growth. As can be seen from Table 7, the moisture content of substrate is negatively correlated to

**Table 7. Pearson correlation analysis of physical and chemical properties of substrate and seedling morphology.**

| | Ground diameter | Seedling height | Total root length | Root surface area | Moisture content | Total porosity before seedling cultivation | Total aeration porosity before seedling cultivation | Organic matter content before seedling cultivation | Total nutrient content before seedling cultivation | Total nutrient after seedling cultivation | Change in organic matter content | Change in total nutrient content |
|---|---|---|---|---|---|---|---|---|---|---|---|---|
| Ground diameter | 1 | 0.982** | 0.044 | 0.181 | -0.972** | -0.891* | -0.438 | -0.723 | 0.754 | 0.723 | -0.905* | 0.732 |
| Seedling height | | 1 | -0.094 | 0.049 | -0.997** | -0.944* | -0.311 | -0.823 | 0.820 | 0.830 | -0.957* | 0.773 |
| Total root length | | | 1 | 0.971** | 0.080 | 0.059 | -0.166 | 0.412 | -0.497 | -0.471 | 0.087 | -0.486 |
| Root surface area | | | | 1 | -0.055 | -0.055 | -0.282 | 0.219 | -0.447 | -0.393 | -0.074 | -0.452 |
| Moisture content | | | | | 1 | 0.967** | 0.238 | 0.818 | -0.826 | -0.853 | 0.966** | -0.771 |
| Total porosity before seedling cultivation | | | | | | 1 | -0.005 | 0.100 | -0.039 | 0.176 | 0.130 | -0.148 |
| Total aeration porosity before seedling cultivation | | | | | | | 1 | 0.838 | -0.814 | -0.937* | 0.952* | -0.710 |
| Organic matter content before seedling cultivation | | | | | | | | 1 | -0.669 | -0.810 | 0.906* | -0.562 |
| Total nutrient content before seedling cultivation | | | | | | | | | 1 | 0.935* | -0.728 | 0.983** |
| Total nutrient after seedling cultivation | | | | | | | | | | 1 | -0.841 | 0.855 |
| Change in organic matter content | | | | | | | | | | | 1 | -0.633 |
| Change in total nutrient content | | | | | | | | | | | | 1 |

* - Correlated at the significant level of 0.05 (bilateral)

** - Correlated at the significant level of 0.01 (bilateral)

seeding ground diameter (r = -0.972, P<0.01) and seedling height (r = -0.997, P<0.01). Seedling height and ground diameter are positively correlated (r = 0.982, P<0.01), and root surface area and root length are positively correlated (r = 0.971, P<0.01). The total porosity of substrate before seedling cultivation is negatively correlated to seedling height (r = -0.891, P<0.05) and ground diameter (r = -0.944, P<0.05). The changes in the contents of organic matter and total nutrient of substrate after seedling cultivation reflect the amounts of substrate nutrients absorbed by the seedlings and the losses of the substrate itself. The change in organic matter content is negatively correlated to ground diameter (r = -0.905, P<0.05) and seedling height (r = -0.957, P<0.05), and is positively correlated to moisture content (r = 0.966, P<0.01), aeration porosity seedling cultivation (r = 0.952, P<0.05) and organic matter content before seedling cultivation (r = 0.906, P<0.05). The change in total nutrient content is positively correlated to the total nutrient content before seedling cultivation (r = 0.983, P<0.01). The total nutrient content after seedling cultivation is negatively correlated to the aeration porosity before seedling cultivation (r = 0.937, P<0.05) and is positively correlated to the total nutrient content before seedling cultivation (r = 0.935, P<0.05). No correlation is found between the change in total nutrient content and seedling ground diameter (r = 0.732) and seedling height (r = 0.773). There are no significant correlations between the contents of organic matter and total nutrient before and after seedling cultivation and seedling ground diameter and height (r1 = -0.723, r2 = -0.823, r3 = 0.304, r4 = 184; r5 = 0.754, r6 = 0.820, r7 = 0.723, and r8 = 0.830). These results suggest that organic matters and nutrients that can be absorbed and utilized by Camellia oleifera seedling play a key role in the seedling growth and development.

## 4. Conclusion

Agricultural and forestry residues were composted using different formulas and the composting efficiencies were evaluated. It is found that the introduction of livestock and poultry manures and the C/N ratios in the range of 25–35 can significantly increase the heating rate, prolong the thermophilic phase, and produce the composts with high total nutrient contents and GIs.

Five compound substrates were prepared using the formulas containing the obtained composts and inorganic substrates, vermiculite and perlite, with no peat added for the cultivation of one year old Camellia oleifera seedlings. The physical and chemical properties of the substrates before and after seedling cultivation and the seedling development were analyzed. The results show that, first, the physical and chemical properties of compost raw materials, composting formula and degree of composting directly affect the physical and chemical properties of the obtained compost and corresponding compound substrate, as well as the seedling growth. Second, the physical and chemical properties of compound substrate conform to the standards defined in GB/T 33891–2017 [16] for seedling substrate, except that their pHs (7.03–7.43) do not meet the pH requirement (5.0–6.5) of the regulation LYT 2314–2014 [31]. However, the Camellia oleifera seedlings developed better on these substrates than on the traditional nursery substrates. Therefore, further study is needed to evaluate the effects of substrate pH on the growth of Camellia oleifera seedling. Third, the survival rates of the seedlings cultivated on the five substrates for 185 day covering the severely hot summer and old winter are all greater than 97.5%. In particular, the seedlings cultivated on substrate C with the highest total nutrient content exhibit 100% survival rate, and the largest seedling height and ground diameter. In addition, all of the seedling heights and 26.67% of ground diameters reach the first grade criteria for two years old seedling. Forth, the changes in the physical and chemical properties of substrate after seedling cultivation are correlated to the growth and development

of Camellia oleifera seedling. The initial total porosity is negatively correlated to seedling height at the significant level, and the change in organic matter content is negatively correlated to ground diameter and seedling height, and positively correlated to the initial aeration porosity and initial organic matter content at significant levels. Based on these results, it can be concluded that the amounts of organic matters and nutrients in the substrate that can be absorbed and utilized by seedling play a key role in the seedling development.

## Supporting information

**S1 Data.**
(XLSX)

## Acknowledgments

The authors would like to thank Changsheng Zhan (Taihu County Huayuan Agricultural Science and Technology Development Co. LTD,Anqing,Anhui, China) for the nursery management and site support.

## Author Contributions

**Conceptualization:** Nianjin Wang.

**Data curation:** Fei Zhou, Jinping Zhang, Lingyan Zhan, Jieman Li.

**Formal analysis:** Fei Zhou, Jinping Zhang.

**Investigation:** Nianjin Wang, Tiantian Zhang, Lingyan Zhan, Jieman Li.

**Methodology:** Jinping Zhang.

**Project administration:** Xiaohua Yao, Xiaofeng Zhang.

**Resources:** Xiaohua Yao, Xiaofeng Zhang.

**Software:** Fei Zhou.

**Writing – original draft:** Fei Zhou.

**Writing – review & editing:** Nianjin Wang, Jinping Zhang.

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
