## [Editor Report · Decision Letter 0]

23 Mar 2022

PONE-D-22-07125Formulation of substrates with agricultural and forestry wastes for Camellia oleifera Abel seedling cultivationPLOS ONE

Dear Dr. Zhang,

Thank you for submitting your manuscript to PLOS ONE. After careful consideration, we feel that it has merit but does not fully meet PLOS ONE’s publication criteria as it currently stands. Therefore, we invite you to submit a revised version of the manuscript that addresses the points raised during the review process.

ACADEMIC EDITOR: 

After perusing the manuscript, I realized you did not follow some of the manuscript formatting structure as required by PLOS ONE. The following were identified as issues as regard submission to the Journal.

No line numbering is seen in the manuscript despite clearly stated in the authors manuscript submission guidelines.The out-of-text or list of references at the end of the manuscript were not handled according the requirement of PLOS ONE, what was done appears to be a different referencing formatting style,There are also few grammatical errors to be corrected.Avoid the use of ‘we’ in the abstract and the paper. Instead use rather “the paper or study ….”The tables and figures were also not handled according to the format. Within the text the positions of tables and figures are supposed to be indicated as “Table 1” and “Fig. 1” and not by inserting the specific tables and figures there. Rather the tables and figures as well as well their respective captions expected to be sent to the end of the manuscript separately.  ==============================

We look forward to receiving your revised manuscript.

Kind regards,

Felix Yao Huemabu Kutsanedzie, Ph.D

Academic Editor

PLOS ONE

Journal Requirements:

"The authors are grateful for the financial support from the National Key R＆D

Program of China (Grant No. 2019YFD1001602) and the Provincial Department of

Science and Technology of Zhejiang, China( Grant NO.2017C02022)."

4. Please include a copy of Table 2 which you refer to in your text.

Additional Editor Comments:

After perusing the manuscript, I realized you did not follow some of the manuscript formatting structure as required by PLOS ONE. The following were identified as issues as regard submission to the Journal.

- No line numbering is seen in the manuscript despite clearly stated in the authors manuscript submission guidelines.

- The out-of-text or list of references at the end of the manuscript were not handled according the requirement of PLOS ONE, what was done appears to be a different referencing formatting style,

- There are also few grammatical errors to be corrected.

- Avoid the use of ‘we’ in the abstract and the paper. Instead use rather “the paper or study ….”

- The tables and figures were also not handled according to the format. Within the text the positions of tables and figures are supposed to be indicated as “Table 1” and “Fig. 1” and not by inserting the specific tables and figures there. Rather the tables and figures as well as well their respective captions

expected to be sent to the end of the manuscript separately.
---

## [Author Response · Author response to Decision Letter 0]

28 Mar 2022

I have read your reply to the manuscript and modified it according to your opinions. The following are the replies one by one.

The whole manuscript has been numbered.

The format of the references has been revised to meet plos ONE requirements.

The grammar of the whole article has been checked and revised in detail, including "we" and "the paper or study....".

The format of tables and figures in the article was also modified, and the tables and pictures were all put at the end of the manuscript. Only "Figure 1" and "Table 1" were left in the position of the charts in the text for illustration.

Table 2 has been reflected in the manuscript.

The new Funding Statement was attached to the cover letter.

---

## [Decision Letter · Decision Letter 1]

6 May 2022

PONE-D-22-07125R1Formulation of substrates with agricultural and forestry wastes for Camellia oleifera Abel seedling cultivationPLOS ONE

Dear Dr. %Jinping Zhang%,

Thank you for submitting your manuscript to PLOS ONE. After careful consideration, we feel that it has merit but does not fully meet PLOS ONE’s publication criteria as it currently stands. Therefore, we invite you to submit a revised version of the manuscript that addresses the points raised during the review process.

ACADEMIC EDITOR: Review comments have been received from reviewers on your manuscript. You are therefore request to answer all queries raised and revise your manuscript for further consideration for publication in PLOS ONE.

We look forward to receiving your revised manuscript.

Kind regards,

Felix Yao Huemabu Kutsanedzie, Ph.D

Academic Editor

PLOS ONE

Journal Requirements:

Additional Editor Comments (if provided):

Review comments have been received from reviewers on your manuscript. You are therefore request to answer all queries raised and revise your manuscript for further consideration for publication in PLOS ONE.

Reviewers' comments:

Reviewer's Responses to Questions

**Comments to the Author**

1. If the authors have adequately addressed your comments raised in a previous round of review and you feel that this manuscript is now acceptable for publication, you may indicate that here to bypass the “Comments to the Author” section, enter your conflict of interest statement in the “Confidential to Editor” section, and submit your "Accept" recommendation.

Reviewer #1: All comments have been addressed

Reviewer #2: All comments have been addressed

2. Is the manuscript technically sound, and do the data support the conclusions?

Reviewer #1: Yes

Reviewer #2: Yes

3. Has the statistical analysis been performed appropriately and rigorously? 

Reviewer #1: Yes

Reviewer #2: Yes

4. Have the authors made all data underlying the findings in their manuscript fully available?

Reviewer #1: Yes

Reviewer #2: Yes

5. Is the manuscript presented in an intelligible fashion and written in standard English?

Reviewer #1: Yes

Reviewer #2: Yes

6. Review Comments to the Author

Reviewer #1: Since all the suggested corrections were incorporated in the manuscript, editor should accept this manuscript in its form

Reviewer #2: 1. The authors satisfactorily addressed the review suggestions proposed in the original submission.

2. Technically, the manuscript is sound and the data provided fully supports the conclusions made. The authors have made all data underlying the findings in their manuscript fully available.

3. Statistical analysis appropriately carried out.

4. The manuscript is presented in standard English. Generally, the manuscript has just a few mechanical inaccuracies. E.g.,

a. Line 20 => should read "... growth based upon..."

b. Line 21 => should read "... volume measurement."

c. Line 25 => should read " reached instead of reach"

d. Line 34 => should read "... after being treated with..."

e. Line 105 => space out "Shell & +"

f. Line 106 => space out "Chip & +"

g. Line 114 - 128 => Justify the paragraphs

5. The references are neither ordered alphabetically nor listed as they appeared in the manuscript yet they have been numbered. The journal format is to be followed appropriately.

7. PLOS authors have the option to publish the peer review history of their article (what does this mean?). If published, this will include your full peer review and any attached files.

Reviewer #1: No

Reviewer #2: **Yes: **Moses Kwaku Golly

---

## [Author Response · Author response to Decision Letter 1]

16 May 2022

I have read your reply to the manuscript and submitted the minimum data set according to your opinion. The following are some minor problems that we found and revised when we summarized the data and made the final accounting, which will be explained here.

In the process of writing the paper, there were some manual errors in the data processing, for example, there were several misread and input errors in the data input, and the standard deviation of part of the mean (fluctuation range) was substituted incorrectly in the calculation, leading to a series of errors in the relevant standard deviation. Fortunately, these errors did not affect the analysis in our paper, and we found them and corrected them one by one before the paper was finalized.

---

## [Editor Report · Decision Letter 2]

25 May 2022

Formulation of substrates with agricultural and forestry wastes for Camellia oleifera Abel seedling cultivation

PONE-D-22-07125R2

Dear Dr. %Jinping Zhang%,

We’re pleased to inform you that your manuscript has been judged scientifically suitable for publication and will be formally accepted for publication once it meets all outstanding technical requirements.

Kind regards,

Felix Yao Huemabu Kutsanedzie, Ph.D

Academic Editor

PLOS ONE

Additional Editor Comments (optional):

Based on your revision of the manuscript in accordance with the suggestions and queries raised by the reviewers of your manuscript, I am glad to inform you at this point of it acceptance for publication consideration in PLOS ONE.
---

## [Editor Report · Acceptance letter]

8 Jul 2022

PONE-D-22-07125R2 

Formulation of substrates with agricultural and forestry wastes for Camellia oleifera Abel seedling cultivation 

Dear Dr. Zhang:

I'm pleased to inform you that your manuscript has been deemed suitable for publication in PLOS ONE. Congratulations! Your manuscript is now with our production department. 

Kind regards, 

on behalf of

Professor Felix Yao Huemabu Kutsanedzie 

Academic Editor

PLOS ONE